# Neighborhood Food Environment and Children’s BMI: A New Framework with Structural Equation Modeling

**DOI:** 10.3390/nu14214631

**Published:** 2022-11-03

**Authors:** Tursunay Abdumijit, Dong Zhao, Ronghua Zhang

**Affiliations:** 1Medical School, Ningbo University, Ningbo 315211, China; 2Department of Nutrition and Food Safety, Zhejiang Provincial Center for Disease Control and Prevention, Hangzhou 310051, China

**Keywords:** food environment, childhood obesity, behaviors

## Abstract

The relationship between neighborhood food environment and childhood obesity is complex and not yet well defined by current research in China, especially when considering the integrated effects with other relative factors. The main purpose of this article is to introduce a framework of children’s weight status, based on their neighborhood food environment, and to identify the impact of food environment on the children’s BMI and potential pathways. The participants of this cross-sectional study were students aged 8–16.5 years old and their parents. Two conceptual frameworks were tested using the structural equation modeling method, and two models were extracted. Model B added the neighborhood food environment based on model A. By comparing the two models, the neighborhood environment was potentially correlated with the children’s BMI directly and may have a positive impact on unhealthy-food eating behaviors, which were positively associated with the children’s BMI. The results suggest that the focus should be placed on the integrated effects of the potential risk factors of childhood obesity, based on considering the neighborhood food environment, which may relate to children’s unhealthy-food eating behaviors and weight status.

## 1. Introduction

An alarming increase in childhood obesity has attracted widespread interest in the fields of public health, nutrition, and psychology, during the past two decades. According to the World Health Organization’s latest report, nearly one-fifth of children and adolescent aged 5 to 18 have suffered from overweight and obesity [1]. In China, this rate has jumped by 17% in the last thirty-five years [2], and it is predicted that roughly one-third of children and adolescents will be overweight and obese by 2030 [3]. It is worth highlighting that the prevalence of childhood obesity differs in different areas due to discrepancies in economic development and the variety of eating habits. For example, some metropolitan areas such as the Jiangsu province, the Zhejiang province, and Shanghai witnessed significant increases in 2017 (31.6%) [4], 2015 (20.2%) [5], and 2014 (31.6%) [6], respectively. Despite much of the research in recent years having focused on identifying potential relative risk factors and practical intervention approaches, this increasing rate is still on an upward trend. This is because of the complexity of the risk factors of childhood obesity. It is generally recognized that genetics, behavior, diet, socioeconomic status, and physical activity are essential influencing aspects of childhood obesity. Among these, with the exception of genetic factors, the factors are believed to be easily modified and managed and have generated considerable research interest in obesity intervention.

Previous studies on childhood obesity modeling have pointed out that parental feeding behaviors [7], parental socioeconomic status [8,9,10], and the child’s food intake [11,12] are considered vital and fundamental potential factors. Moreover, nutritional education has been considered the most practical and financially friendly method to tackle the obesity pandemic around the world. A growing number of earlier works demonstrated that nutritional knowledge has a positive impact on reducing childhood obesity [13,14]. However, there is no research reporting the integrated effects of these factors on children’s weight in China. Consequently, it is difficult to understand the correlation between the potential risk factors and children’s weight status.

In addition, in recent years, several studies have revealed that environmental factors potentially increase the prevalence of obesity among children and adults, as these factors strongly impact people’s lifestyles [15,16,17,18]. Theoretically, an unhealthy neighborhood food environment leads to unhealthy-food eating. For example, the number of fast-food restaurants is significantly correlated to the fast-food intake of residents [19]. Similarly, the number of supermarket/convenience stores may increase access to unhealthy snacks, which can be directly associated with gain weight [20]. Unfortunately, there is little evidence reported from China regarding this. The research by Zhou M et al. [21] and the research by Zhang M et al. [22] reported that the food environment has an impact on obesity among middle-aged and older adults in China. However, there is little evidence for this among children and adolescents. Zhou PL and his colleagues discovered that Western-style fast-food is positively correlated with adolescents’ body mass index (BMI), while Chinese-style fast-food is negatively associated with the BMI of adolescents. They also pointed out that the positive impacts of Chinese-style fast-food and fresh vegetables and fruits have gradually disappeared in recent years [23]. Furthermore, no attention has been paid to the integrating effect of a healthy- or unhealthy-food environment on children’s weight status. Consequently, it is still unclear how neighborhood food environment impacts children’s food intake and weight. Fortunately, the structural equation modeling (SEM) method can be used to clarify the casual, direct, and indirect relationship between the latent and dependent variables [24].

Therefore, to bridge this gap, this study aimed to introduce two integrated models, applying a combination of four basic concepts (family socioeconomic status, neighborhood food environment, parental feeding behaviors, and children’s unhealthy-food intake) and the nutritional knowledge level of both the children and their parents (or caregivers). Additionally, these factors were compared to identify the relationship between neighborhood food environment and the children’s BMI.

## 2. Materials and Methods

### 2.1. Research Framework

To identify the effect of a child’s neighborhood environment on their BMI, we designed two different frameworks. Framework A presented the logical relationship between the family’s socioeconomic status, parental feeding behaviors, children’s unhealthy-food eating behaviors, nutritional knowledge level, and the children’s BMI. Framework B illustrated the additional effect of the child’s neighborhood food environment on their BMI, based on framework A. Family socioeconomic status was an independent variable, and the neighborhood food environment, parental feeding behaviors, children’s unhealthy-food eating behaviors, and parent’s and children’s nutrition knowledge levels were four mediators. The children’s BMI was the dependent variable. The research framework is demonstrated in Figure 1.

### 2.2. Participants

The present study included 4970 children aged 8–16.5 years old, as well as their parents, from 36 schools participating in the National Surveillance of Nutritional Health Status of Rural Students in Compulsory Education project. This is a community-based children’s health survey that was conducted from October to December in 2021 in the coastal–rural area of Zhejiang, China. Fornell and his colleagues suggested that the minimum sample size for SEM analysis highly depends on the complexity of the model and the measurements of the model characteristics [25]. Consequently, this study required at least 300 participants, according to the characteristics of our theoretical model (Appendix A).

### 2.3. Measurement of Variables

As the dependent variable, the children’s BMI was calculated using the following formula: BMI = weight(kg)/[height(m)]^2^. Their weight was measured by trained investigators, using the Jian min Weight Scale (GMCS-I Type) (manufactured in China), and rounded to 0.1 kg. Their height was measured twice, with the Jian min Height Scale (GMCS-I Type) (manufactured in China), and recorded in meters (m), with an accuracy of 0.1 m. Each student’s weight and height were measured twice, and the average values were used to calculate the child’s BMI. According to the Chinese diagnostic method for overweight and obesity among school-aged children and adolescents, WS/T-2018 [26], the children’s weight was classified into four categories: underweight, normal, overweight, and obese (Table 1).

### 2.4. Family Socioeconomic Status

A family’ socioeconomic status was based on the family’s income level and parental education level. The families’ incomes were classified into six groups: “less than CNY 20,000/year”, “CNY 20,000–49,000/year”, ”CNY 50,000–99,000/year”, “CNY 100,000–199,000/year”, and “CNY ≥200,000/year” (CNY 10,000 = USD1389.7768). With respect to education level, the responses obtained were categorized as “less than primary school”, “elementary school”, “secondary school”, “high school”, “Bachelor’s”, and “Master’s and Ph.D.”.

### 2.5. Neighborhood Food Environment

The neighborhood food environment was assessed based on the number of fast-food restaurants, Chinese-style restaurants, fruit and vegetable stores, supermarkets/convenience stores, and milk tea shops/bakeries/dessert shops. The neighborhood was defined as being within the scope of 500 m from the resident’s home.

### 2.6. Children’s Unhealthy-Food Eating Behaviors

The frequency of eating snacks, sugary beverages, and deep-fried foods was categorized as “less than 1 time/week”, “1∼2 times/week”, “3∼4 times/week”, “5∼6 times/week”, “1 time/day” and “≥2 times/day”. The respondents were asked about the money spent on buying snacks, and the responses were denoted as “less than CNY 1/day”, “CNY 1∼3.9/day”, “CNY 4∼6.9/day”, “CNY 7∼9.9/day”, and “more than CNY 10/day”. (CNY 1 = USD 0.138).

### 2.7. Parental Feeding Behaviors and Nutritional Knowledge Level

The parental feeding behaviors only considered the behaviors that might be directly associated with the children’s unhealthy-food eating. These included buying snacks and sugar-sweetened drinks for the child. Parents indicated their responses on a four-point scale (“never”, “1∼2 times/month”, “3∼4 times/month” and “more than 2 times/week”), regarding how often they buy certain foods for their children. The nutritional knowledge level was measured based on the 10 standard questions. Appendix B presents the standard questions used for measuring the parents’ and children’s nutritional knowledge level.

### 2.8. Statistical Analysis

Proportions were calculated as percentages of the categorical variables. The multiple imputation method was used for the estimation of missing values in the SAS 9.4 software (the missing values included 1023 data from 14 variables, and the missing rate ranged from 1.32 to 13.9%).

Latent variables, such as family socioeconomic status, neighborhood food environment, parental feeding behaviors, and children’s unhealthy-food eating behaviors, were identified using exploratory factor analysis (EFA) with 14 nutrition questionnaire items. EFA and Cronbach’s α test were undertaken using the IBM SPSS Statistics Version 23.0 software. The Kaiser–Meyer–Olkin and Bartlett’s tests of sphericity were used to evaluate the data adequacy and suitability. A Kaiser–Meyer–Olkin value of >0.5 and a Bartlett’s test of sphericity with a *p* value of <0.05 indicated that the data were qualified for use in the principal component analysis [27]. To determine the number of factors, the Kaiser’s criterion (eigenvalues > 1.0), the screen test, and parallel analysis were used [28]. The items with absolute factor loadings > 0.3 were retained [27]. To determine the internal consistency, the ordinal α coefficient (ordinal equivalent of Cronbach’s α) was estimated [27], for which values greater than 0.6 were considered as acceptable [25]. Furthermore, the average variance extracted (AVE) was used to test the reliability of the study questionnaire; the AVE values of each latent variable should be equal or higher than 0.5.

After the EFA, confirmatory factor analysis was conducted to evaluate the measurement model. In addition, pathway analysis was used to test the hypothesized direct and indirect relationships between the latent variables and dependent variables [29], using the IBM AMOSTM Version 23.0 software(IBM (International Business Machines Corporation) and SPSS company, New York, NY, USA). In SEM, maximum likelihood is the foundational estimation technique required, as the data must meet the normality standard. Thus, a normality analysis was conducted. To evaluate the models’ fits, the following goodness-of-fit indices were assessed: Chi-square, the goodness-of-fit index (GFI), the comparative fit index (CFI) the Tucker–Lewis index [TLI], and the root mean square standard error of approximation (RMSEA). For GFI, CFI, and TLI, values greater than 0.9 implied an acceptable fit, and values greater than 0.95 were considered a good fit [30]. For RMSEA, values smaller than 0.08 indicated an acceptable fit, and values smaller than 0.06 meant a good fit. In the SEM method, the direct and indirect effects can be estimated, and, if their values were 0.1, 0.3, and 0.5, they were considered small, medium, and strong effects, respectively [31].

## 3. Results

### 3.1. Descriptive Statistical Analysis

This study consisted of 4970 students from 18 primary and 18 secondary schools (the third to ninth grades), which included 52.70% boys and 47.30% girls in the coastal–rural areas of the Zhejiang province. The number of primary school students accounted for 56.34% of the children who participated, and the number of students from secondary schools was 43.66%. The average age of students was approximately 11.8 years old, and their average weight was 43.98 Kg. Concerning the children’s weight status, 6.62% of the children were underweight, 72.60% of the children were normal weight, and 27.41% of the children were overweight and obese (overweight: 15.86% and obese: 11.55%) (Table 1).

In terms of the neighborhood food environment, more than half of the students responded that there were no fast-food restaurants in their neighborhood, while merely 2.2% of the students reported that there were more than five fast-food restaurants local to them. Similarly, the proportion of students living in areas without bakeries/milk tea shop/cafés in the neighborhood was 41.95%; however, for the students who lived in areas with more bakeries/milk tea shops/cafés, this figure was only 6.22%. A total of 15.88% of the students ate snacks daily; however, the figure for students who drank sugar-sweetened beverages was only about 4.35%. Regarding family conditions, most of the parent’s educational levels were at secondary school and high school levels (fathers: 68.25% and mothers: 65.49%). Moreover, 64.95% of the students’ families’ yearly income was between CNY 50,000–199,000. In terms of parental feeding, 88.85% and 90.42% of the parents bought snacks and sugary beverages for their children less than 2 times/month, respectively (Appendix A).

### 3.2. SEM Analysis

#### Validity and Reliability

In terms of the study’s validity, the Cronbach’s alpha of the four latent variables ranged from 0.601 to 0.864, and all of them met the acceptable criteria of internal consistency. Additionally, in terms of the reliability of the latent constructs, with the exception of the children’s unhealthy-food eating behaviors, the other construct values were higher than 0.5 and considered to be acceptable (Table 2).

### 3.3. Exploratory Factor Analysis

Table 3 presents the results of the EFA conducted on 14 items, which extracted four factors with eigenvalues > 1.0 and explained 61.09% of the total variance. The Kaiser–Meyer–Olkin value (0.768) indicated sampling adequacy, and Bartlett’s test of sphericity (χ^2^ = 19393.89, *p* < 0.0001) satisfied the required criteria (*p* < 0.05) and supported the factorability of the dataset. The items with absolute factor loadings > 0.3 were retained.

### 3.4. The Model Fit Analysis

The values of the model A fit indices were as follows: GFI = 0.986, TLI = 0.946, and CFI = 0.960, The model’s goodness-of-fit was accepted (χ^2^/df = 5.706, *p* < 0.001, RMSEA = 0.022, 90% CI: 0.020–0.023). The values of the model B fit indices were as follows: GFI = 0.98, TLI = 0.966, and CFI = 0.972. The model’s goodness-of-fit was accepted (χ^2^/df = 4.480, *p* < 0.001, RMSEA = 0.019, 90% CI: 0.018–0.020).

### 3.5. Structural Model

To identify the effect of the neighborhood food environment on children’s BMI and potential pathways, two structural models were applied in this study; they are presented in Figure 2 and Figure 3.

The data shown in Figure 2 revealed that family socioeconomic status has a positive impact on parents’ nutritional knowledge levels and parental feeding behaviors, while it has a negative influence on children’s unhealthy-food eating behaviors and children’s BMI. Similarly, the parents’ nutritional knowledge level was positively associated with the children’s nutritional knowledge, which was negatively correlated with the unhealthy-food eating behaviors of children. However, a converse relationship also appeared between the parental feeding behaviors and the children’s BMI. Moreover, children’s unhealthy-food eating behaviors significantly increased the children’s BMI. Medium effects can be seen between the parental feeding behaviors and the children’s unhealthy-food eating behaviors, as well as between the family’s socioeconomic status and the parents’ nutritional knowledge level.

Figure 3 shows that the neighborhood food environment was related to the children’s BMI in direct and indirect ways. Two pathways were mapped out: in the first pathway, the neighborhood food environment was a mediator between the family’s socioeconomic status and the unhealthy-food eating habits of children; it is evident that the socioeconomic status was positively associated with the neighborhood food environment, which has a positive impact on the children’s unhealthy eating behaviors. The second pathway demonstrated the direct and positive relationship between the neighborhood food environment and the children’s BMI. The relationship between the other constructs was similar to that of the results shown in Figure 2.

The indirect effect of the family’s socioeconomic status, in regard to parents’ nutrition knowledge, significantly decreased the children’s BMI (model A: −0.0016, *p* < 0.001; model B: −0.0015, *p* < 0.001). Moreover, the indirect effect of the family’s socioeconomic status, in terms of parental feeding behaviors, also decreased the children’s BMI (model A: −0.0056, *p* < 0.001; model B: −0.0056, *p* < 0.001). In addition, the indirect effect of the family’s socioeconomic status, through the children’s unhealthy-food eating behaviors, reduced the children’s BMI. (model A: −0.014, *p* < 0.001; model B: −0.014, *p* < 0.001). However, the indirect effect of the family’s socioeconomic status, through parental feeding behaviors and the children’s unhealthy-food eating behaviors, increased the children’s BMI (model A: 0.0034, *p* < 0.001; model B: 0.0031, *p* < 0.001). The indirect effect of the family’s socioeconomic status, in terms of the neighborhood environment, significantly increased the children’s BMI (model B: 0.0076, *p* < 0.001). Similar results can also be seen in the indirect effect of the family’s socioeconomic status via the neighborhood environment and the children’s unhealthy-food eating behaviors (model B: 0.0076, 0.0001, *p* < 0.001). Moreover, the indirect effect of parental feeding behaviors, in terms of the children’s unhealthy-food eating, dramatically increased the children’s BMI (model A: 0.042, *p* < 0.001; model B: 0.0385, *p* < 0.001). With the exception of the indirect impacts, the parents’ nutritional knowledge, in regard to the children’s nutritional knowledge levels and the children’s unhealthy-food eating behaviors, was found to have a positive impact on reducing the children’s BMI in both models (model A: −0.0054, *p* < 0.001; model B: −0.0049, *p* < 0.001). Additionally, the indirect effect of the children’s nutritional knowledge levels, in terms of the children’s unhealthy-food eating behaviors, was also correlated with a decrease in the children’s BMI (model A: −0.0192, *p* < 0.001; model B: −0.0176, *p* < 0.001).

## 4. Discussion

The total prevalence rate of overweight and obesity was 27.23%, which is considerably higher than the national level (19% in 2020) [32] and the international level (20% in 2016) [1], and lower than the prevalence rate among children and adolescents in the Jiangsu province, China (33.2% in 2019–2020) [33]; Shanghai, China (30.31% in 2019) [34]; Maya, Mexico (39%) [35]; and Thrace, NE Greece (48.5% in 2018) [36]. This study examined a conceptual and multi-factorial model and offered an understanding of the intricate relationship between children’s neighborhood food environment and their BMI by applying SEM. The children’s BMI was the primary dependent variable, and the family’s socioeconomic status was the primary independent variable. Two types of models were extracted from our data, based on our hypothesized model, which had a good fit. A direct and positive relationship can be seen between the family’s socioeconomic status and the children’s BMI, as well as between the parental feeding behaviors and the children’s BMI. Conversely, a direct and negative relationship can be seen between the neighborhood food environment and the children’s BMI, as well as between the unhealthy-food eating behaviors and the children’s BMI.

Four latent constructs were extracted, using the exploratory factor analysis, and their validity and reliability were evaluated according to the value of the average variance extracted (AVE), Cronbach’s alpha, and composite reliabilities (CR). The data indicated that, with the exception of the AVE value of the children’s unhealthy-food eating behaviors, the other latent constructs met the acceptable value standards. However, Fornell and D. F. Larcker suggested that if the AVE is between 0.4 to 0.5, and the composite reliabilities of the latent constructs are higher than 0.6, this indicates that the convergent validity of the construct is still adequate [25]. Therefore, the validity and reliability of the four latent variables of this study met the acceptable criteria.

According to the structural models, in model A, the conceptual relationship between the family’s economic status, parental feeding behaviors, the children’s unhealthy-food eating behaviors, the nutritional knowledge level, and the children’s BMI were demonstrated to be statistically significant. Family socioeconomic status was inversely associated with the children’s weight status in this study. This suggested that parents with higher incomes and higher educational levels are more likely to have a positive influence on their children’s weight control. Similar results have been documented in previous research, conducted by Huang H et al. [10], Crouch, P et al. [37], and Sares-Jaske L et al. [38]. Moreover, the data in our study revealed that a high family socioeconomic status has a positive impact on the parents’ nutritional knowledge levels, which are directly associated with the children’s nutritional knowledge levels in both models. It is logical that a parent’s higher educational background may lead to obtaining a higher socioeconomic status and a higher level of nutritional knowledge. Furthermore, the parents with a higher level of nutritional knowledge were more likely to teach their children more regarding nutritional education. Simultaneously, they might also guide their children toward healthy eating and help them to form good eating habits. This finding was in line with the previous literature on childhood obesity [39,40,41], except that parental feeding behaviors were found to significantly increase children’s unhealthy-food intake in both models. This is consistent with the results obtained in the study by Gevers DWM et al., who found that parental feeding behaviors can contribute to the specific food intakes of children by controlling and monitoring children’s eating behaviors [42]. For example, unhealthy parental feeding behaviors, such as a lack of monitoring [43], or over-restricting [44], the children’s unhealthy-food intake, have detrimental impacts on child weight control. Therefore, this is evidence that reasonable parental monitoring is more likely to lead to children having healthier eating habits. Regarding children’s unhealthy-food eating behaviors, such as a high consumption of fast-food, sugar-sweetened beverages, and deep-fried foods (e.g., chips, fried chicken, fried noodles, and so on), are the main risk factor for childhood obesity [44,45,46,47,48]. Similarly, this study demonstrated that children’s unhealthy-food eating behaviors are a crucial mediator between the other three latent constructs and the children’s BMI.

In comparison to model A, the data in model B indicated that the neighborhood food environment may have a positive impact on children’s BMI via two pathways: (1) the direct association between the neighborhood food environment and children’s BMI; (2) indirectly, via mediation of the children’s unhealthy-food eating behaviors. This finding was consistent with the study by Shier V et al., which demonstrated that greater exposure to fast-food restaurants, supermarkets, and convenience stores was correlated with a higher BMI among youth in the USA [49]. Moreover, a systematic review conducted in China reported that the neighborhood food environment had an impact on the diet and obesity of the residents [50]. Similar evidence from South Korea reported that unhealthy behaviors of children were linked to the neighborhood environment, and a greater availability of fast-food restaurants and convenience stores was more likely positively associated with increases in children’s BMI, among the children who had obesity [51]. It is worth mentioning that the integrated impact was positive, but not considered strong. The neighborhood food environment in this study not only included unhealthy-food environments, such Western fast-food restaurants, convenience stores/supermarkets, and bakeries/cafés/milk tea shops, but also included healthy-food environments, such as Chinese-style fast-food restaurants and fruit and vegetable stores. In this model, the family’s socioeconomic status had a positive impact on the neighborhood environment. A possible explanation is that the family’s socioeconomic status plays a key role in the neighborhood’s socioeconomic condition, which was recognized as a vital determinant of obesity prevalence among the Chinese population [22]. Therefore, it can be concluded that the total impact of the neighborhood food environment on the participants was more likely to increase the children’s BMI.

## 5. Limitations

Due to the cross-sectional nature of this study, it was difficult to determine the temporal relationships. We suggest that the model of the current study could be tested using longitudinal data, which may provide researchers with clearer and more accurate causal relationships. Moreover, there were limitations in the data collection; we did not collect data regarding neighborhood security, physical activity, sleep time, and built environments, which are deemed remarkable factors in causing childhood obesity. Therefore, it is recommended that these factors are studied in future investigations. Additionally, the present study focused only on the proximal environment; however, the macro environment, including socioeconomic conditions, political policies, and the neighborhood environment of schools, was not measured in the research framework used in this study. We also recommend that future studies take these factors into consideration.

## 6. Conclusions

This study identified the intricate relationship between the neighborhood food environment and children’s BMI. The findings provide a foundation for our follow-up intervention study, which will be conducted next year. The findings of this study suggest that the focus should be placed on the integrated effects of the potential risk factors of childhood obesity, based on considering the neighborhood food environment, which may be related to children’s unhealthy-food eating behaviors and weight status. This study provides some evidence for the childhood obesity model, based on the neighborhood food environment.

## Figures and Tables

**Figure 1 nutrients-14-04631-f001:**
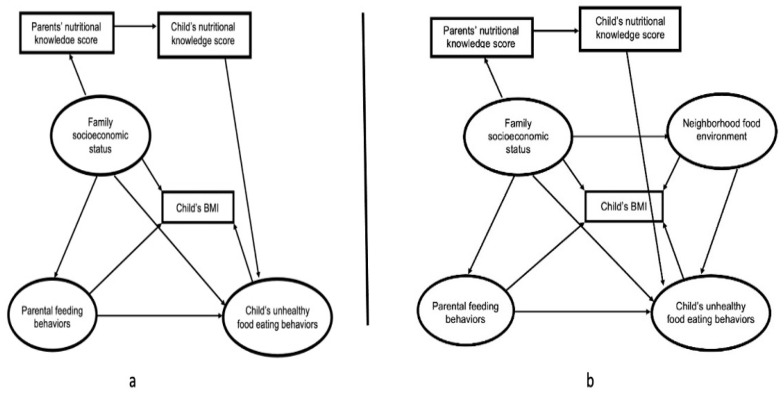
Research framework. Framework **a**: the model demonstrated the integrated effects of family, parent, and children’s characteristics on the children’s BMI.; Framework **b**: the model included the potential impact of the neighborhood food environment based on framework a on the children’s BMI.Child’s BMI was calculated by this formula: BMI = weight(kg)/[height(m)]^2^).

**Figure 2 nutrients-14-04631-f002:**
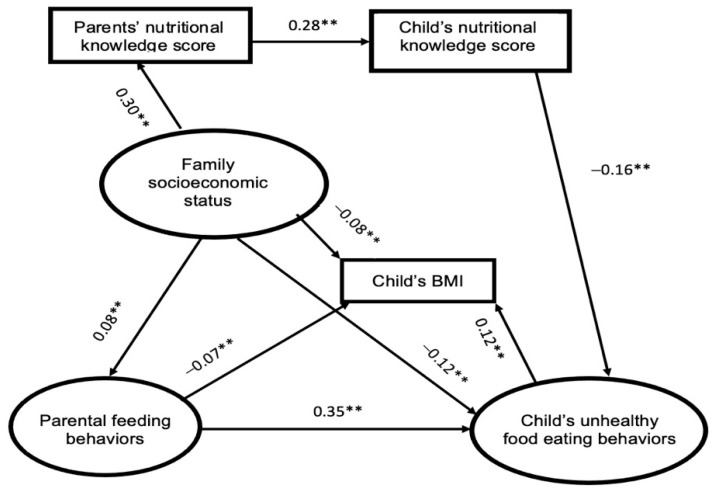
Model A (the solid line represents statistically significant values. ** *p* < 0.0001).

**Figure 3 nutrients-14-04631-f003:**
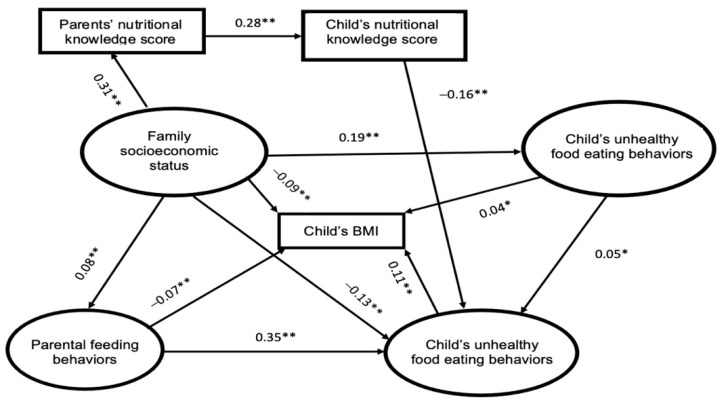
Model B (the solid line represents statistically significant values. * *p* < 0.05; ** *p* < 0.0001).

**Table 1 nutrients-14-04631-t001:** The socio-demographic characteristics of the children participants.

	Number	Percentage/Mean ± SD
Gender		
Boy	2619	52.7
Girl	2351	47.3
School grade		
Third	698	14.04
Fourth	722	14.53
Fifth	707	14.23
Sixth	673	13.54
Seventh	748	15.05
Eighth	685	13.78
Ninth	737	14.83
School type		
Primary school	2800	56.34
Secondary school	2170	43.66
Children’s age	4970	11.84 ± 2.03
Children’s weight	4970	43.98 ± 13.43
Children’s BMI	4970	19.07 ± 3.55
Weight status		
Underweight	324	6.52
Normal	3284	72.6
Overweight	788	15.86
Obese	574	11.55
Children’s nutritional knowledge score	4970	5.16 ± 1.83
Parent’s nutritional knowledge score	4970	6.51 ± 1.57

**Table 2 nutrients-14-04631-t002:** The average variance extracted (AVE), Cronbach’s alpha, and composite reliabilities of the latent constructs in the model of neighborhood food environment correlates of childhood and adolescent obesity.

Latent Constructs	AVE	Cronbach’s Alpha	CR
Family socioeconomic status	0.619	0.696	0.827
Parental feeding behaviors	0.725	0.653	0.841
Child’s unhealthy-food eating behaviors	0.465	0.601	0.775
Neighborhood food environment	0.646	0.864	0.901

**Table 3 nutrients-14-04631-t003:** Exploratory factor analysis with Varimax rotation and internal consistency for factors and their respective items.

Factors/Factor Items	Factor Loading	Variance Explained (%)
Neighborhood food environment		23.25
Fast-food restaurants	0.743	
Chinese-style restaurants	0.831	
Fruit and vegetable stores	0.845	
Supermarket/convenience stores	0.752	
Milk tea shops/bakeries/dessert shops/cafés	0.844	
Family socioeconomic status		13.69
Father’s education	0.854	
Mother’s education	0.852	
Family income	0.634	
Parental feeding		10.55
Snacks	0.861	
Sugar-sweetened beverages	0.842	
Children’s unhealthy-food intake		13.60
Snacks	0.603	
Money for buying snacks	0.638	
Sugar-sweetened beverages	0.752	
Deep-fried food	0.724	

## Data Availability

The data supporting the findings of the study are available from the corresponding author on reasonable request.

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
