# Peer review of "Neighborhood Food Environment and Children’s BMI: A New Framework with Structural Equation Modeling"

_nutrients, 2022, doi:10.3390/nu14214631_

Round 1

Reviewer 1 Report

Nutrientes- 1988868 COMENTARIOS DE REVISIÓN

COMENTARIOS

Gracias a los autores por su envío a Nutrients. Este manuscrito investigó el desarrollo de un marco de referencia para la obesidad infantil basado en el entorno alimentario del vecindario e identificó correlatos dietéticos de la obesidad infantil. Reconozco plenamente el tiempo y el esfuerzo dedicados al análisis de los resultados y la posterior elaboración de un manuscrito. Sin embargo, el documento necesita ser revisado.

RESUMEN

The abstract presented by the authors conforms to the requirements and facilitates the understanding of the paper.

INTRODUCTION

The authors present a concise and well-written introduction section and include all the necessary information to contextualise and justify the need for this study.

MATERIALS AND METHODS

The methodology used in the development of the study is described in a simple and complete way to understand the research process.

RESULTS

Bibliographical references are included in the section "Validity and Reliability", in the "Multicollinearity Analysis" and in the "Structural Model". These references should be cited in the discussion section and not in the results section. The implications of the results should ideally be developed in the discussion section. The results section should only describe the data obtained.

DISCUSSION

The discussion section details the main findings of the study. I suggest including the limitations of the study in this section.

CONCLUSIONS

El apartado de conclusiones debe ser claro y conciso, y debe responder al objetivo propuesto. Sugiero evitar "Los resultados sugieren...". Asimismo, las posibles implicaciones de los resultados obtenidos cobran más sentido en el apartado de Discusión. Considere esto y modifique esta sección en consecuencia.

En general, las líneas del manuscrito deben estar numeradas para facilitar la revisión y rápida identificación de posibles modificaciones.

Atentamente,

-

COMMENTS Thanks to the authors for their submission to Nutrients. This manuscript investigated the development of a childhood obesity framework based on the neighbourhood food environment and identified dietary correlates of childhood obesity. I fully acknowledge the time and effort spent in the analysis of the results and the subsequent development of a manuscript. However, the paper needs to be revised. ABSTRACT The abstract presented by the authors conforms to the requirements and facilitates the understanding of the paper. INTRODUCTION The authors present a concise and well-written introduction section and include all the necessary information to contextualise and justify the need for this study. MATERIALS AND METHODS The methodology used in the development of the study is described in a simple and complete way to understand the research process. RESULTS Bibliographical references are included in the section "Validity and Reliability", in the "Multicollinearity Analysis" and in the "Structural Model". These references should be cited in the discussion section and not in the results section. The implications of the results should ideally be developed in the discussion section. The results section should only describe the data obtained. DISCUSSION The discussion section details the main findings of the study. I suggest including the limitations of the study in this section. CONCLUSIONS The conclusions section should be clear and concise, and should answer the proposed objective. I suggest avoiding "The results suggest ...". Also, the possible implications of the results obtained make more sense in the Discussion section. Please consider this and modify this section accordingly. In general, the lines of the manuscript should be numbered to facilitate review and quick identification of possible modifications. 

Reviewer 2 Report

This paper provides analysis as a framework for childhood obesity based on neighborhood food environment and socioeconomic factors. While I find the topic to be very important, I found the presentation of the background, methods, and results to be difficult to get through because so much is presented. Narrowing the focus would improve this paper dramatically. Please find more details below.

1. There are grammatical errors and structure issues throughout the paper that make it difficult to understand key take home points. 

2. I feel like the purpose of the paper was to create a framework to work off of, however, from how the paper is currently written it seems more like an exercise in use of SEM using this type of data rather than creating a structure to work off of in obesity.

3. I find the placement of the research framework in figure 1 in the introduction oddly placed - seems like it would be better placed in the method- analysis section to describe the analytical approach that will be taken. Moreover, I am a little confused by it since the introduction and title led me to believe that the focus was on neighborhoods but the framework presents it as a component of a larger model.

4. I think it would be helpful to provide more details on the date - purpose, who was include, how does it represent or not represent children in China.

5. The first paragraph on page 3 confuses me and I don't think is necessary, or better placed in the analysis description.

6. Same comment for Table 1.

7. Table 2 takes up a lot of space and it seems like the information was covered better in the text.

8. On page 5 in the statistical analysis section. The authors need to better describe the multiple imputation procedures- how much data was missing and for how many variables. Extensive imputing could lead to unintended autocorrelations and inflated robustness of associations. 

9. The authors provide extensive description of their methodology on page 5 and 6 without any rationale on why it is being conducted and how will it answer the question as to how neighborhood are related to obesity in children. 

10. There are so many variables being tested here beyond neighborhood that results are hard to follow. I believe there is adequate data at the neighborhood level to conduct multilevel models to assess the relationship between neighborhoods and childhood obesity- which would be an important story to tell.

11.  For an international audience it would be helpful to explain the Chinese monetary system. 

12. Table 5 should come up further in the paper. Seems like that should be one of the first things discussed is the prevalence of childhood obesity.

13. Seems like table 7 is the key story to your analysis but it is buried in with the other tables and figures and as a result neighborhood effects become secondary.

14. I am not sure what the purpose of Figure 3 is.

15. There are two figure 3's both seem to be unnecessary providing little to the storyline of this paper.

16. Same comment for Table 8.

17. Figure 4 seems to have a lot of extra that does not really help us understand the relationship between childhood obesity and neighborhood food environment- there are no direct pathways from parents' or childrens' knowledge and obesity and really the relationship that mapped out is questionable since knowledge and eating behaviors cancel each other out.

18. Figure 5 tells me that neighborhoods do not have any direct relationship with childhood obesity. Is that correct?

19. If point 18 is correct then the conclusions on page 15 are unsubstantiated with the analysis presented. 
